# Comprehensive Kinase Activity Profiling Revealed the Kinase Activity Patterns Associated with the Effects of EGFR Tyrosine Kinase Inhibitor Therapy in Advanced Non-Small-Cell Lung Cancer Patients with Sensitizing EGFR Mutations

**DOI:** 10.3390/proteomes11010006

**Published:** 2023-02-05

**Authors:** Rei Noguchi, Akihiro Yoshimura, Junji Uchino, Takayuki Takeda, Yusuke Chihara, Takayo Ota, Osamu Hiranuma, Hiroshi Gyotoku, Koichi Takayama, Tadashi Kondo

**Affiliations:** 1Division of Rare Cancer Research, National Cancer Center Research Institute, Tokyo 104-0045, Japan; 2Department of Pulmonary Medicine, Graduate School of Medical Science, Kyoto Prefectural University of Medicine, Kyoto 602-8566, Japan; 3Bannan Central Hospital, Shizuoka 438-0814, Japan; 4Department of Respiratory Medicine, Japanese Red Cross Kyoto Daini Hospital, Kyoto 602-8026, Japan; 5Department of Respiratory Medicine, Uji-Tokushukai Medical Center, Kyoto 611-0041, Japan; 6Department of Medical Oncology, Izumi City General Hospital, Osaka 594-0073, Japan; 7Department of Respiratory Medicine, Otsu City Hospital, Shiga 520-0804, Japan; 8Department of Respiratory Medicine, Graduate School of Biomedical Sciences, Nagasaki University, Nagasaki 852-8501, Japan

**Keywords:** non-small-cell lung cancer, sensitizing EGFR mutation, kinase activity, prognosis, kinome

## Abstract

EGFR mutations are strong predictive markers for EGFR tyrosine kinase inhibitor (EGFR-TKI) therapy in patients with non-small-cell lung cancer (NSCLC). Although NSCLC patients with sensitizing EGFR mutations have better prognoses, some patients exhibit worse prognoses. We hypothesized that various activities of kinases could be potential predictive biomarkers for EGFR-TKI treatment among NSCLC patients with sensitizing EGFR mutations. In 18 patients with stage IV NSCLC, EGFR mutations were detected and comprehensive kinase activity profiling was performed using the peptide array PamStation12 for 100 tyrosine kinases. Prognoses were observed prospectively after the administration of EGFR-TKIs. Finally, the kinase profiles were analyzed in combination with the prognoses of the patients. Comprehensive kinase activity analysis identified specific kinase features, consisting of 102 peptides and 35 kinases, in NSCLC patients with sensitizing EGFR mutations. Network analysis revealed seven highly phosphorylated kinases: CTNNB1, CRK, EGFR, ERBB2, PIK3R1, PLCG1, and PTPN11. Pathway analysis and Reactome analysis revealed that the PI3K-AKT and RAF/ MAPK pathways were significantly enriched in the poor prognosis group, being consistent with the outcome of the network analysis. Patients with poor prognoses exhibited high activation of EGFR, PIK3R1, and ERBB2. Comprehensive kinase activity profiles may provide predictive biomarker candidates for screening patients with advanced NSCLC harboring sensitizing EGFR mutations.

## 1. Introduction

Lung cancer is one of the most common cancers, causing death frequently [1]. Among lung cancers, non-small-cell lung cancer (NSCLC) amounts to approximately 80% of newly diagnosed lung cancers yearly [2]. Notably, most NSCLC cases are diagnosed as lung cancers with advanced stage [2]. For patients with advanced NSCLC, chemotherapy has been considered the front-line remedy. However, patients are provided with limited therapeutic effects and poor clinical outcomes due to its toxicity and adverse events: the outcomes are a median overall survival (OS) of only 8–10 months and a 5-year survival rate of less than 15% [3]. Epidermal growth factor receptor tyrosine kinase inhibitors (EGFR-TKIs) are used to treat patients with advanced or metastatic NSCLC. EGFR-TKIs have demonstrated a significant effect on NSCLC patients harboring EGFR mutations and have improved quality of life [4,5].

The overexpression of EGFR is commonly identified in NSCLC (32–81%) and is known as a reliable target and biomarker for NSCLC treatment [6,7]. EGFR-TKIs including gefitinib and erlotinib demonstrate surpassing clinical effects compared to standard chemotherapy or best supportive care [4,5]. Previous studies have reported that EGFR mutation is a practical predictive marker of increased sensitivity to EGFR-TKIs and concerned with the improvement of progression-free survival with TKIs [8]. Particularly, deletions of exon 19 and point mutations of exon 21 (21-L858R) are commonly observed in 85% of patients with NSCLC harboring EGFR mutations [9,10,11]. These mutations demonstrate a high response rate of 70% against EGFR-TKI [12]. Several studies have confirmed that patients with NSCLC harboring these two mutations show fewer side effects and improved quality of life [13]. However, the heterogeneity of responses to EGFR TKIs has been pointed out to have a wide range from a few months to several years in progression-free survival. Additionally, in around 10–20% of patients with sensitizing *EGFR* mutations, objective responses to EGFR TKIs are not exhibited. These phenomena have motivated the search for other predictive biomarkers surpassing EGFR mutations that can detect patients with sensitizing EGFR mutations who have a worse prognosis.

Protein phosphorylation is an essential apparatus regulating cellular functions such as apoptosis, cell proliferation and migration, cell cycle, and differentiation [14]. Approximately 500 different kinases orchestrating these pivotal functions are encoded by the human genome [15], and 90% of all proteins are subjected to phosphorylation [16]. Aberrated kinase activity is caused by genetic mutations such as amplification, point mutation, chromosomal translocation, and epigenetic regulation in carcinogenesis and cancer progression. Furthermore, the dysregulation of self-phosphorylation and the kinase-to-kinase regulatory relationship also result in the aberrant activity of kinases. The corrupted kinase activity finally causes various effects including the interruption of important cell functions, the transformation of normal cells into tumor cells, and the determination of malignant features such as invasion, metastasis, and resistance to chemotherapy [17]. Therefore, aberrantly regulated kinases and their substrate proteins are considered biomarkers to affect the process of cancer treatments [18,19]. In addition, protein kinases are crucial therapeutic targets in oncology [20], and a number of approved kinase inhibitors have been used for cancer therapeutics [21]. Therefore, the research of protein phosphorylation will deepen the understanding of fundamental biology and provide novel insights for clinical applications of NSCLC harboring sensitizing mutations.

Although most NSCLC patients with sensitizing EGFR mutations exhibit a better prognosis, some exhibit a worse prognosis. For the identification of predictive biomarkers that are useful to detect these patients with worse prognoses, we performed a comprehensive kinase activity analysis using PamStation 12 (PamGene International, BJ’s-Hertogenbosch, The Netherlands) in NSCLC patients at advanced stages with sensitizing EGFR mutations and followed their prognosis prospectively. Finally, kinomic profiles may provide prognostic biomarkers for patients with advanced NSCLC harboring sensitizing EGFR mutations.

## 2. Materials and Methods

### 2.1. Patients

Nineteen patients with advanced NSCLC who were treated in the enrolled hospitals between June 2018 and October 2020 were included in the study. All patients had available tumor tissues for biomarker analysis. Tumor tissue from the primary sites was obtained by surgery or biopsy. The samples were rinsed with saline after being taken. The samples were simply put into a −80 °C freezer. The time between biopsy and freezing was conducted within a few minutes in all samples. Serial sections were used for mutational and tyrosine kinase activity analyses. After the diagnosis of NSCLC, EGFR-TKIs were administered as monotherapy (composed of the first therapy in gefitinib, afatinib, or osimertinib) and complete clinicopathological findings were examined in all patients. Treatment with gefitinib (250 mg), afatinib (20–40 mg), or osimertinib (80 mg) alone was maintained unless disease progression, adverse events, or patient refusal occurred. Written informed consent was obtained from all patients. The study protocol was approved by the Institutional Ethics Committee of the National Cancer Center (2018-208) and the Kyoto Prefectural University of Medicine (ERB-C-1106).

### 2.2. Study Design

This prospective observational study was designed to evaluate tyrosine kinase activity for predicting the clinical response to EGFR-TKI treatment in patients with advanced NSCLC harboring activating EGFR mutations. Tumor samples were obtained at the initial diagnosis. Until all clinical data were evaluated, the clinical data were enclosed during laboratory analysis Recorded clinical data consisted of age, sex, smoking history, pathology, stage at diagnosis, treatments, EGFR mutation, and adverse events. After the administration of EGFR-TKIs, the prognosis was followed. Efficacy evaluations included the best response, disease control rate (DCR), objective response rate (ORR), progression-free survival (PFS), and overall survival (OS). The outline of this study is presented in Figure 1.

### 2.3. Assessments

Tumors were assessed at diagnosis and every 8–12 weeks until the investigators reported disease progression or unacceptable adverse events. According to the Response Evaluation Criteria in Solid Tumors (RECIST), clinical responses to TKIs consisting of complete response (CR), partial response (PR), stable disease (SD), and disease progression (PD) were examined [22]. The definition of PFS was the time from the initiation of TKI treatment to PD or death. The definition of OS was the time from TKI initiation to death. All films were assessed by an independent radiologist who was blinded to the EGFR biomarker status.

### 2.4. EGFR-mutation Analysis

EGFR mutations in exons 18–21 were examined using the polymerase chain reaction method for tumor and plasma samples. Sequencing was performed at commercial clinical laboratories (SRL, Inc., Tokyo, Japan).

### 2.5. Comprehensive Tyrosine Kinase Activity Assay

Frozen biopsy tissues were processed at 4 °C for kinomic profiling after grinding. The tissues were lysed in M-PER Mammalian Extraction Buffer (Pierce, Rockford, IL, USA) with Halt protease and phosphatase inhibitors (Pierce cat. 78420, 78415) [23]. Protein quantification was performed using a Bradford reaction assay. The extracted protein (5 mg) was mixed with kinase buffer, ATP, and fluorescently labeled anti-PY20 antibodies. Then, the mixed protein was loaded into the tyrosine (PTK) PamChips. The samples were subjected to the PamStation 12 kinomics workstation (PamGene International, BJ’s-Hertogenbosch, The Netherlands) using the PTK PamChip protocol in Evolve12 Software (v. 1.5) (PamGene International), as previously reported [23]. When lysates were pumped through the array, images were captured, analyzed, and quantified using BioNavigator v. 5.1 (PamGene International). The study was performed in duplicate and subjected to activity analysis.

### 2.6. Identification of Upstream Kinases

Comprehensive kinase activity was analyzed using BioNavigator software v. 6.3.67.0 (PamGene International). The analysis of kinase activity was conducted corresponding to the phosphorylated peptide-specific reaction per each “spot” on the PamChip. The intensities of the raw signals of the 144 spots were measured over multiple 50 ms exposures sequentially as lysates were pumped through the array. The measurement was also performed over multiple exposure times (10, 20, 50, 100, and 200 ms) after the lysates were rinsed off. These values were converted to slopes of intensity by exposure time. The slopes were multiplied by 100 and log2-transformed. The unsupervised hierarchical clustering of the kinomic activity data was performed using Euclidean distance. Active kinases were predicted using PhosphoSitePlus (https://www.phosphosite.org, accessed on 25 January 2023) [24] and the UniProt database (https://www.uniprot.org/, accessed on 25 January 2023) [25]. These kinases, which were found in all the databases, were considered positively identified.

### 2.7. STRING Analysis

The Search Tool for the Retrieval of Interacting Genes/Proteins (STRING) database (https://string-db.org/, accessed on 25 January 2023) was used to examine the protein–protein interaction (PPI) network. The STRING app in Cytoscape (https://cytoscape.org/index.html, accessed on 1 February 2023)was used to examine the potential correlation between these signal intensities and each peptide phosphorylation [26]. To perform the PPI network analysis, significant peptides were selected (*p*-value using a *t*-test between cluster 1 and cluster 2). These peptides (*n* = 2) were translated into their corresponding UniProt ID using UniProt (https://www.uniprot.org/, accessed on 25 January 2023). After the removal of duplicated UniProt IDs, UniProt IDs and fold changes were entered into the STRING database. In the setting for organism, “Homo sapiens” was selected.

### 2.8. Pathway Analysis, Network Analysis, and Reactome Analysis

STRING analysis was also used for pathway analysis. The dataset after conversion to UniProt ID was entered as an input into STRING. UniProt IDs were mapped onto pathways based on curated data from the literature. The top 20 pathways were grouped according to the processes. The identified peptides were used for the network analysis. The Uniprot IDs corresponding to the peptides were used in Cytoscape software version 3.9.3 [27] and the STRING database [28]. The network analysis was conducted using a network analyzer. Hub bottlenecks were determined following the degree values and betweenness centrality. Common hub bottlenecks and top nodes were identified as the central nodes according to closeness centrality and stress. Action maps consisting of activation, inhibition, and expression were demonstrated for the central nodes using CluePedia [29]. The Kapa score was considered to be the default value in CluePedia. Reactome analysis was conducted with the ReactomePA package with the *p*-value cutoff set as 0.05 [30]. Only the top entries with a minimum adjusted *p*-value were included in the dotplot.

### 2.9. Statistical Analysis

Progression-free survival (PFS) was calculated from the date of EGFR-TKI administration to the date of disease progression or death from any cause, and overall survival (OS) was calculated from the date of EGFR-TKI administration to the date of death from any cause. Fisher’s exact test or χ2 test for categorical variables and the *t*-test for continuous variables were used to analyze the clinicopathological features of the two groups divided by the cut-off value. Survival curves were plotted using the Kaplan–Meier method and compared using the log-rank test. All statistical analyses were performed using the GraphPad Prism software (v.9.0; GraphPad Software, San Diego, CA, USA).

## 3. Results

### 3.1. Patient Characteristics

Patient characteristics are summarized in Table 1. The median age was 73.0 years (range, 46–88 years). There were six men and thirteen women, of whom six were non-smokers and thirteen were smokers. All patients had stage IV adenocarcinoma. A total of 11 cases with exon 19-del mutation and 8 cases with exon 21 L858R mutation were identified. As EGFR-TKI therapy, two patients received first-line therapy as gefitinib, three patients as afatinib, and fourteen patients as osimertinib. One case was not evaluated using a comprehensive kinase activity assay because the amount of extracted protein did not meet the required amount for the experiment. Thus, 18 cases were analyzed. Based on RECIST 1.1, fifteen patients were categorized as PR, one patient was categorized as SD and PD, and two patients were not evaluated as NE. Based on the results of the RECIST, the objective response rate (ORR) was 88.24% and the disease control rate (DCR) was 94.12%.

### 3.2. Comprehensive Kinase Activity Analysis in NSCLC Patients with Sensitizing EGFR Mutations

The tyrosine kinase activity profiles of advanced NSCLC patients with sensitizing EGFR mutations were examined in nineteen NSCLC patients with pathologically confirmed NSCLC who underwent panel sequencing. In one patient, the amount of protein was too low to measure the protein kinase activity. Therefore, this sample was excluded from the analysis. Except for this case, all NSCLC tissue samples demonstrated protein kinase activity profiles. As a quality control, peptides that showed no increase in signal intensity over time were eliminated. After the quality control, 102 of 144 PTK peptides were evaluated. The values of mean signal intensity per peptide were calculated and log2-transformed (Appendix A). In all samples, the kinase activity assay was performed in duplicate.

### 3.3. Identification of Peptides Showing Significant Activation of Phosphorylation in Advanced NSCLC with Sensitizing EGFR Mutations

The results were imaged as a heat map (Figure 2a); the rows of the heatmap represent each peptide, and the columns of the heatmap represent each sample. The peptides were grouped according to the signal intensity of NSCLC patients with sensitizing EGFR mutations (Figure 2a). The peptides were classified into three groups: cluster A, intermediately phosphorylated group; cluster B, highly phosphorylated group; and cluster C, poorly phosphorylated group (Figure 2a). These included peptides in cluster A, 34 peptides; cluster B, 29 peptides; and cluster C, 39 peptides (Appendix A). The samples were classified into two groups: cluster 1, the lower phosphorylated group, and cluster 2, the higher phosphorylated group (Figure 2a). Cluster 1 included 19 samples and cluster 2 comprised 17 samples.

### 3.4. Putative Upstream Kinases

Of these 102 peptides, 29 peptides were identified to be able to phosphorylate the tyrosines (upstream kinases). The putative tyrosine kinases for each peptide are listed in Table 2, Appendix A, and Figure 2b. Six receptor tyrosine kinases (RTK), ERRB2, FGFR2, MET, PDGFRB, RET, and VEGFR2, were identified, as well as members of the four non-receptor tyrosine kinases (NRTK), including Abl1, Lck, SRC, and Syk in cluster A, the intermediately phosphorylated group. In the highly phosphorylated group, cluster B, the two RTKs, PDGFRB and VEGFR2, the three NRTKs, FER, FES, and Syk, and the serine/threonine kinase, WEE1, were identified. The six RTKs, including EGFR, EPHA4, ERBB4, FGFR1, INSR, and VEGFR2, the two NRTKs, including ABL1 and SRC, and the five serine/threonine kinases (STKs), including MAP2K1, MAP2K2, MAP2K3, MAP2K4, and MAP2K6, were poorly phosphorylated in cluster C (Figure 2b). Based on the Venn diagram, one kinase, VEGFR2, was phosphorylated in all groups. Except for VEGFR2, two kinases were commonly identified in the two clusters: PDGFRB and SYK were detected both in clusters A and B, and SRC and ABL1 were detected both in clusters A and C.

### 3.5. Pathway Analysis and Network Analysis

Pathway analysis (STRING) using the UniProt IDs of the 102 peptides and the signal intensity values for each peptide yielded many pathways (Table 3). The identified pathways are highly significant with the *p*-values ranging from 4.27 × 10–23 to 2.21 × 10–12. The significantly enriched pathways were the PI3K-Akt, Ras, Rap1, and MAPK signaling pathways.

Network analysis revealed that a network including 102 UniProt IDs was constructed (Figure 3a). The network contained seven isolated nodes and a main connected component (78 nodes and 619 edges). The top 10% of nodes based on degree values, including CTNNB1, EGFR, PIK3R1, ERBB2, PTPN11, PLCG1, CRK, and CBL, were selected as hubs. The top 10 nodes regarding betweenness centrality, CTNNB1, EGFR, PIK3R1, PLCG1, ERBB2, MAPK1, PTPN11, and CRK, were determined as bottlenecks. Common hubs and bottlenecks, including CTNNB1, EGFR, PIK3R1, ERBB2, PTPN11, PLCG1, and CRK, were identified (Table 4 and Appendix A). The action map of the identified seven kinases is shown in Figure 3b.

### 3.6. Kinase Profile Different between Highly Phosphorylated and Lower Phosphorylated Group

In the previous heat map, cases were grouped according to the signal intensity of NSCLC patients with sensitizing EGFR mutations (Figure 2a). The cases were classified into two groups: cluster 1, the lower phosphorylated group, and cluster 2, the highly phosphorylated group. The 19 samples from the 10 cases were classified into cluster 1, and the other 19 samples from the other 10 cases were classified into cluster 2 (Appendix A). Peptides showing significant differences in phosphorylation between clusters 1 and 2 were identified. Thirty-five peptides were identified as differentially phosphorylated (*p* < 0.05, FDR = 0.090) (Appendix A). For these 35 peptides, kinases reported to be able to phosphorylate tyrosine as upstream kinases were identified. The putative tyrosine kinases upstream of each peptide are listed in Table 5. Several receptor tyrosine kinases such as EGFR and JAK2 have been identified.

### 3.7. Pathway Analysis, Reactome Analysis, and Network Analysis between Highly Phosphorylated Group and Low Phosphorylated Group

Pathway analysis using the UniProt IDs of the 35 peptides and the value of signal intensities for each peptide yielded many pathways (Table 6). These pathways were highly significant, as the *p*-values range from 2.74 × 10^–23^ to 1.10 × 10^–12^. The 20 most significant pathways were enriched in the PI3K-Akt, Ras, Rap1, and MAPK signaling pathways, which were similar to those of NSCLC patients with sensitizing EGFR mutations.

Reactome analysis using the UniProt IDs of the 29 peptides with *p*-values less than 0.05 in the *t*-test yielded many pathways (Figure 4). The pathways, including the PI3K-AKT and RAF/MAP pathways, were involved in the highly phosphorylated group.

Network analysis revealed that a network consisting of 35 UniProt IDs was constructed (Figure 5). The network contained three isolated nodes and a main connected component (26 nodes and 93 edges). The top 10% of the nodes based on degree values, including EGFR, PIK3R1, and ERBB2, were selected as hubs. The top ten nodes regarding betweenness centrality contained EGFR, PIK3R1, and ERBB2 as bottlenecks. Common hubs and bottlenecks, including EGFR, PIK3R1, and ERBB2, were identified. All hub bottlenecks were included in the top nodes based on closeness centrality and stress (Table 7 and Appendix A).

### 3.8. Disease Free Survival (DFS) and Overall Survival (OS)

The relative hazard for local progression or death between patients with cluster 1 and cluster 2 was 0.5119 (95% CI, 0.1149–2.281; *p* = 0.38), with a median duration of 20.7 months vs 39.4 months (Figure 6a). Survival between the study groups did not significantly improve in cluster 2 (HR: 0.7359; 95% CI, 0.1270–4.263; *p* = 0.732) (Figure 6b).

## 4. Discussion

The present study investigated comprehensive kinomic profiles to identify commonly upregulated kinases and develop prognostic markers for patients with NSCLC harboring sensitizing EGFR mutations. This is the first study to demonstrate the activity of multiple tyrosine kinases in NSCLC patients harboring sensitizing EGFR mutations. Samples from 18 NSCLC patients with EGFR mutations were used to perform kinase activity analysis using PamStation12. This was a prospective study that integrated kinase information and clinical outcomes. Notably, prominent tyrosine kinases of the ErbB receptor family, EGFR and ERBB2, and kinases belonging to downstream signaling pathways, including CRK, CTNNB1, PIK3R1, PLCG1, and PTPN11, were highly activated in all patients. In particular, tyrosine receptor kinases, including EGFR, ERBB2, and PIK3R1, represent key components that affect patient prognosis. Pathway analysis and Reactome analysis revealed that PI3K-AKT and RAF/MAPK signaling pathways were enriched in NSCLC with sensitizing EGFR mutations. 

### 4.1. Common Activated Kinases in NSCLC with Sensitizing EGFR Mutations

Our study revealed that seven kinases, CTNNB1, CRK, EGFR, ERBB2, PIK3R1, PLCG1, and PTPN11, were highly activated in all NSCLC patients harboring sensitizing EGFR mutations. Among these seven kinases, CTNNB1, CRK, PLCG1, and PTPN11 were newly identified as highly activated kinases in our study. Β-catenin, encoded by the CTNNB1 gene, plays an important role in a signaling pathway affecting cell proliferation and differentiation [31]. Despite the low frequency of mutations in CTNNB1 in NSCLC, CTNNB1 contributes to the development of NSCLC through EGFR mutations in vitro and in vivo from Nakamura et al. [32]. The CT10 regulator of kinase (CRK) is a family of widely expressed adaptor proteins involved in signal transduction from various oncoproteins, including Bcr-Abl, EGFR, PDGF, and VEGFR [33,34] and plays essential roles in cytoskeletal changes, cell proliferation, adhesion, and migration [35]. The overexpression of CRK has been reported in NSCLC [36] and the high phosphorylation of CRK has been verified in NSCLC cell lines [37]. Phospholipase C gamma 1 (PLCG1) is a subtype of phospholipase C gamma (PLCg), a lipase activated by receptors in the cellular membrane including RTKs and adhesion receptors [38]. Wenqiang et al. reported that using in vitro and in vivo models of NSCLC and the phosphorylation of PLCG1 promote tumor growth in NSCLC, which is consistent with our results [38]. PTPN11, protein tyrosine phosphatase non-receptor type 11, is a member of the protein tyrosine phosphatase (PTP) family [39] and regulates several molecules involved in Ras signaling [40]. Mutations in PTPN11 have been linked to the pathogenesis of leukemia and breast cancer. However, a low prevalence of somatic PTPN11 mutations has been detected in lung cancer [41]. In addition to the prevalence of PTPN11 mutations, our results suggest that the phosphorylation of PTPN11 is related to NSCLC harboring sensitizing EGFR mutations. Altogether, these newly identified kinases with high activation are important for deepening the pathogenesis of NSCLC with sensitizing EGFR mutations.

### 4.2. Activated Kinases in NSCLC Patients with Poor Prognosis

This study revealed that three kinases, EGFR, ERBB2, and PIK3R1, were prognostic biomarkers because they were detected as highly activated kinases in the patient group with poor prognosis. The overexpression of EGFR is commonly observed in NSCLC patients (40–80%) [42,43] and is associated with poor prognosis [44,45,46]. HER2 is a member of the erbB transmembrane receptor family. Increased HER2 expression has an association with inferior survival in patients with NSCLC, and high EGFR and HER2 co-expression has an additive impact on unfavorable prognosis [47]. Additionally, Rikova et al. reported that NSCLC tumors express highly phosphorylated EGFR and ERBB2 at above-average concentrations [48], which was concordant with our results. Increased PI3K/Akt activity has also been observed in NSCLC. However, the finding that p-Akt has no association with EGFR-TKI efficacy is conflicting [49]. PIK3R1 encodes the regulatory subunit (p85a) of PIK3CA. Activating mutations in PIK3R1 have been reported in several cancers, including colon cancer and glioblastoma, and lead to the activation of the PI3K-AKT pathway [49]. Our study identified PIK3R1 phosphorylation as a prognostic biomarker in patients with NSCLC with sensitizing EGFR mutations. In addition, we also elucidated that the co-occurrence of EGFR, HER2, and PIK3R1 phosphorylation was associated with poor prognosis in patients with NSCLC. Our gene ontology analysis and Reactome analysis demonstrated that the PI3K-AKT and RAF/MAPK signaling pathways were enriched in the group with poor prognosis, which supports newly identified kinases as predictive biomarkers. These observations strongly suggested the utility of the kinase profiling approach to the prognostic biomarker development, and warrant further validation studies using the additional samples.

### 4.3. Limitations

Our study had several limitations. First, the number of patients with a worse prognosis was too small compared with those with a favorable prognosis. Ideally, an equal number of patients with both prognoses should be included in the data-mining analysis. Secondly, since most of the enrolled patients did not have any adverse events, we could not determine whether the peptides and kinases were significantly associated with the adverse events. Third, three samples in cluster 2 seemed to be more similar to those in cluster 1. We examined the clinical features of those samples, but we could not identify the factor that would make an appropriate explanation for the obvious discordance. Fourth, many confounding factors such as the different efficacies of osimertinib vs other EGFR TKIs, the site of metastases, tumor burden, and performance status have not been taken into consideration. These limitations stemmed from the limited number of cases, and we could solve them by including more patients. Toward clinical applications, further investigation should be required to identify the predictive biomarkers with statistical significance. Fifth, the EGFR undergoes other posttranslational modifications which are not only Y phosphorylation but also S phosphorylation as proteoforms. In the current study, a single modification has been evaluated and the addition of other modification data will deepen our understanding of the proteoforms in advanced NSCLC patients harboring sensitive EGFR mutations. The current molecular targeted drugs were largely targeting Y phosphorylation, and we employed the PamChip which allows the investigation of only Y phosphorylation. The PamChip for S phosphorylation will further our understanding of the kinome backgrounds of cancer progression. Sixth, the STRING and Reactome depend on the literature review, and the kinase functions that were not reported yet were not considered for the interpretation. Thus, multiple interpretations other than those in this discussion will be possible in the near future. Overall, our analysis strongly suggested the utility of kinase profiling for the development of predictive biomarkers and warrants further investigation. 

## 5. Conclusions

Comprehensive kinase activity analysis using 18 samples derived from patients with NSCLC harboring sensitizing EGFR mutations identified common kinomic profiles and kinases that are specific to patients with poor prognoses. We need more samples from different patients and more experiments using different methods to obtain conclusive experimental results. Continuous and collaborative efforts will thus be required to identify prognostic markers using comprehensive kinomic profiles.

## Figures and Tables

**Figure 1 proteomes-11-00006-f001:**
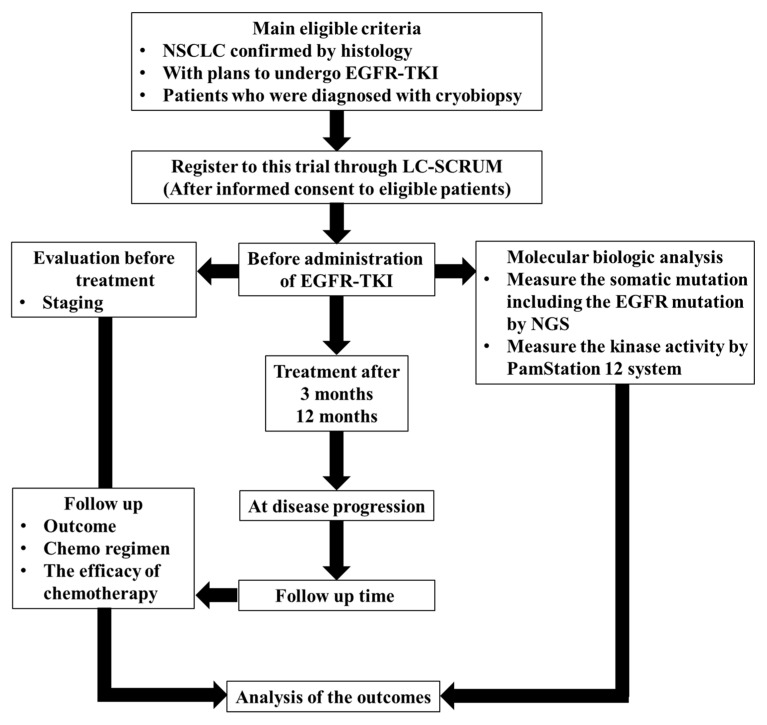
Workflow of this study. A number of NSCLC patients with sensitizing EGFR mutations were enrolled in this study. The prognosis was followed.

**Figure 2 proteomes-11-00006-f002:**
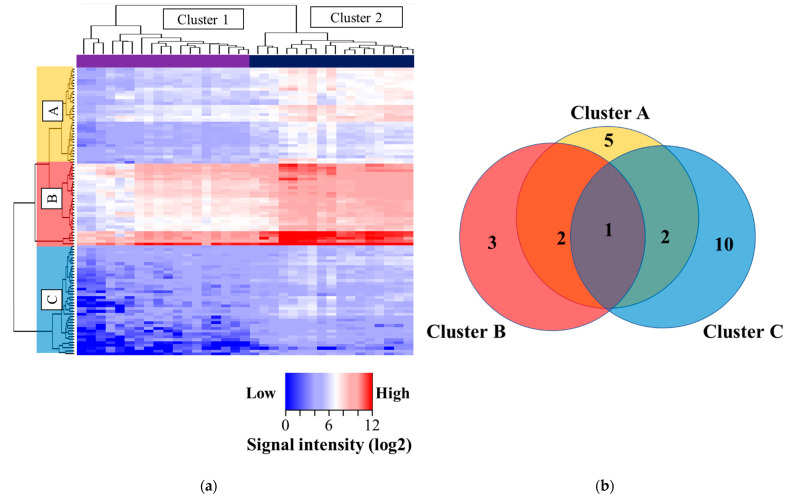
Kinase activity profiles in advanced patients with NSCLC with sensitizing EGFR mutations. (**a**) Kinase activity was comprehensively analyzed using clustering and heatmap analyses. The rows represent peptides and the columns represent samples. The peptides were categorized into three groups according to their kinase activity: cluster A, intermediate group (yellow); cluster B, highly activated group (red); and cluster C, poorly activated group (blue). The right column represents the expected kinase types according to the corresponding peptide sites. The samples were categorized into two groups according to their kinase activity: cluster 1, poorly phosphorylated group (purple), and cluster 2, highly phosphorylated group (navy). (**b**) Venn diagram of identified kinases among the three clusters.

**Figure 3 proteomes-11-00006-f003:**
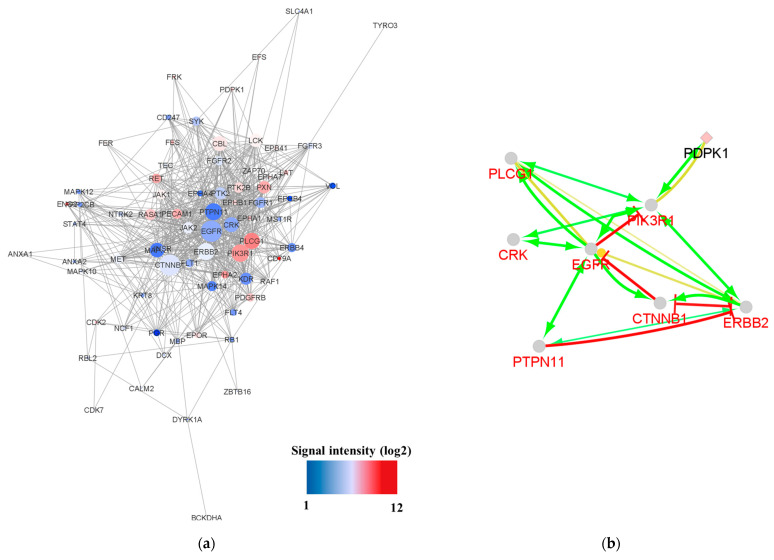
Network analysis of kinase activity in advanced patients with NSCLC with sensitizing EGFR mutations. (**a**) Altered kinases from each patient were mapped on a network using the Cytoscape STRING application. Red nodes represent higher kinase activity and blue nodes represent lower kinase activity. The size of the node represents the number of connections between nodes. (**b**) Action map of the identified 7 kinases. Binding (blue), activation (green), inhibition (red), catalysis (purple), reaction (black), expression (yellow), and ptmod (pink) actions for the 7 kinases. The map was illustrated using the CluePedia-STRING ACTIONS-v10.5-20.11.2017 program.

**Figure 4 proteomes-11-00006-f004:**
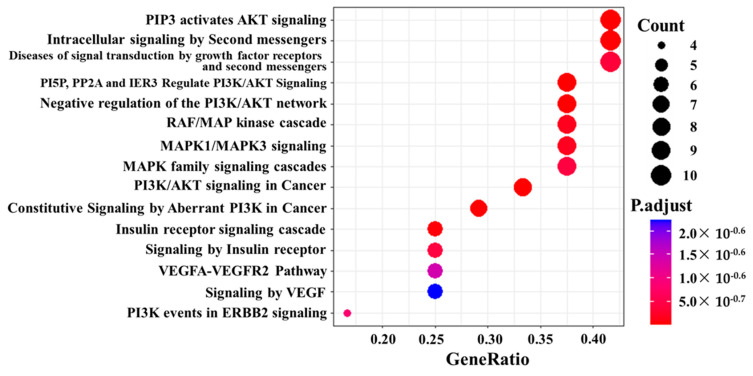
Reactome analysis of kinase activity in comparison between cluster 1 and cluster 2. Dot size describes the number of genes; gene ratio is defined as the percentage of genes in each pathway compared with all genes in the samples.

**Figure 5 proteomes-11-00006-f005:**
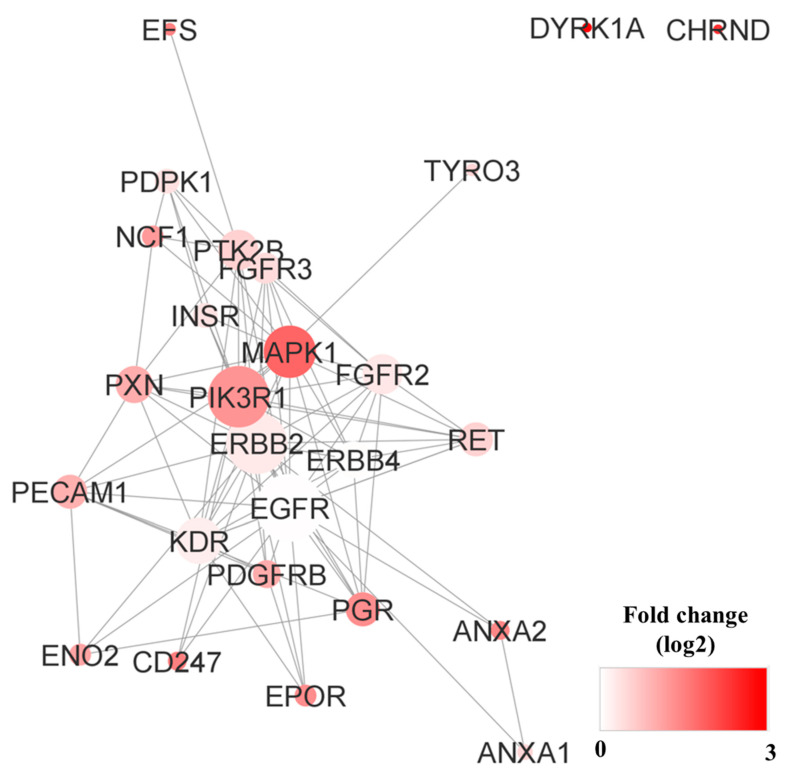
Network analysis of kinase activity in poor prognosis. Altered kinases from each patient were mapped onto a network using the Cytoscape STRING app. Red nodes represent higher fold change in kinase activity, and white nodes represent lower fold change in kinase activity. The size of a node represents the number of connections between the nodes.

**Figure 6 proteomes-11-00006-f006:**
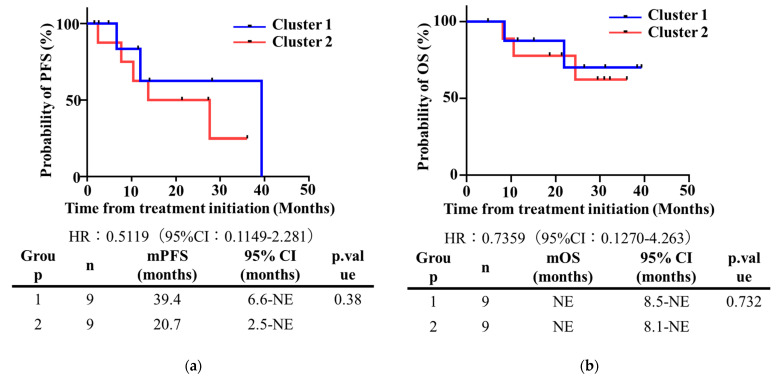
Kaplan-Meier survival curves. Disease-free survival (**a**) and overall survival (**b**) in clusters 1 and 2.

**Table 1 proteomes-11-00006-t001:** Patient characteristics.

Characteristics		*n* (%)
Median age, years		73.0 (46.0–88.0)
Age	<70	7 (36.8)
	≧70	12 (63.2)
Gender	Male	6 (31.6)
	Female	13 (68.4)
Smoking history	Never	6 (31.6)
	Current or former	13 (68.4)
ECOG performance status	0	6 (31.6)
	1	13 (68.4)
	2	2 (10.5)
Histology	Adenocarcinoma	18 (94.7)
	NOS	1 (5.3)
Stage	IV	19 (100)
EGFR mutation	Del19	11 (57.9)
	L858R	8 (42.1)
EGFR TKI therapy	Gefitinib	2 (10.5)
	Afatinib	3 (15.8)
	Osimertinib	14 (73.7)
Response	PR	15 (78.9)
	SD	1 (5.3)
	PD	1 (5.3)
	NE	2 (10.5)
ORR		88.24%
DCR		94.12%
Median MTS		44.1 (−36.0–80.0)
Proteome analysis	Cluster 1	9 (47.4)
	Cluster 2	9 (47.4)
	Unanalyzable	1 (5.3)

Abbreviations: EGFR, epidermal growth factor receptor; PR, partial response; SD, stable disease; PD, progressive disease; NE, not evaluated; ORR, objective response rate; DCR, disease control rate; TKI, tyrosine kinase.

**Table 2 proteomes-11-00006-t002:** List of kinases based on clustering analysis.

Peptide ID	Uniprot Accession	Protein Name	Sequence	Tyr	Type of Cluster	UniProt	Phosphosite Plus
PGFRB_1002_1014	P09619	PDGFRB	LDTSSVLYTAVQP	(1009)	Cluster A	**PDGFRB**	**PDGFRB**
MET_1227_1239	P08581	MET	RDMYDKEYYSVHN	(1230, 1234, 1235)	Cluster A	**MET**	**MET**, Ron
RAF1_332_344	P04049	RAF1	PRGQRDSSYYWEI	(340, 341)	Cluster A	**SRC**	**SRC**
ERBB2_1241_1253	P04626	ERBB2	PTAENPEYLGLDV	(1248)	Cluster A	**ERBB2**	**ERBB2**
FGFR2_762_774	P21802	FGFR2	TLTTNEEYLDLSQ	(769)	Cluster A	**FGFR2**	**FGFR2**
LCK_387_399	P06239	LCK	RLIEDNEYTAREG	(394)	Cluster A	**Lck**	**Lck**, AXL, yopH
PDPK1_369_381	O15530	PDPK1	DEDCYGNYDNLLS	(373, 376)	Cluster A	**SRC**, INSR	**SRC**
CBL_693_705	P22681	CBL	EGEEDTEYMTPSS	(700)	Cluster A	**ABl1**	**Abl**, Fyn, INSR
FAK1_569_581	Q05397	PTK2	RYMEDSTYYKASK	(570, 576, 577)	Cluster A	**RET**, SRC	FAK, FGR, Met, **RET**
PGFRB_771_783	P09619	PDGFRB	YMAPYDNYVPSAP	(771, 775, 778)	Cluster A	**PDGFRB**	**PDGFRB**
KSYK_518_530	P43405	SYK	ALRADENYYKAQT	(525, 526)	Cluster A	**SYK**	**SYK**, Lyn
PGFRB_768_780	P09619	PDGFRB	SSNYMAPYDNYVP	(771, 775, 778)	Cluster A	**PDGFRB**	**PDGFRB**
VGFR2_1168_1180	P35968	KDR	AQQDGKDYIVLPI	(1175)	Cluster A	**VEGFR2**	Src, **VEGFR2**
VGFR2_1052_1064	P35968	KDR	DIYKDPDYVRKGD	(1054, 1059)	Cluster A	**VEGFR2**	**VEGFR2**
PLCG1_764_776	P19174	PLCG1	IGTAEPDYGALYE	(771, 775)	Cluster B	**SYK**	Abl, EGFR, **SYK**
PAXI_111_123	P49023	PXN	VGEEEHVYSFPNK	(118)	Cluster B	PTK6	Abl, FAK
FES_706_718	P07332	FES	REEADGVYAASGG	(713)	Cluster B	**FES**	**FES**
PGFRB_572_584	P09619	PDGFRB	VSSDGHEYIYVDP	(579, 581)	Cluster B	**PDGFRB**	**PDGFRB**
CDK2_8_20	P24941	CDK2	EKIGEGTYGVVYK	(15, 19)	Cluster B	**WEE1**	**WEE1**
VGFR2_989_1001	P35968	KDR	EEAPEDLYKDFLT	(996)	Cluster B	**VEGFR2**	**VEGFR2**
PGFRB_1014_1028	P09619	PDGFRB	PNEGDNDYIIPLPDP	(1021)	Cluster B	**PDGFRB**	**PDGFRB**
FER_707_719	P16591	FER	RQEDGGVYSSSGL	(714)	Cluster B	**FER**	**FER**, Src
VGFR2_1046_1058	P35968	KDR	DFGLARDIYKDPD	(1054)	Cluster C	**VEGFR2**	**VEGFR2**
INSR_992_1004	P06213	INSR	YASSNPEYLSASD	(992, 999)	Cluster C	**INSR**	**INSR**
EGFR_1165_1177	P00533	EGFR	ISLDNPDYQQDFF	(1172)	Cluster C	**EGFR**	**EGFR**
ERBB4_1277_1289	Q15303	ERBB4	IVAENPEYLSEFS	(1284)	Cluster C	**ERBB4**	**HER4**
EGFR_1190_1202	P00533	EGFR	STAENAEYLRVAP	(1197)	Cluster C	**EGFR**	**EGFR**
FGFR1_761_773	P11362	FGFR1	TSNQEYLDLSMPL	(766)	Cluster C	**FGFR1**	**FGFR1**
CRK_214_226	P46108	CRK	GPPEPGPYAQPSV	(221)	Cluster C	**ABL1**	**ABL1**
VGFR3_1061_1073	P35916	FLT4	DIYKDPDYVRKGS	(1063, 1068)	Cluster C	**SRC**, FLT4	**SRC**
ANXA2_17_29	P07355	ANXA2	HSTPPSAYGSVKA	(24)	Cluster C	**SRC**	**SRC**, IGF1R, Yes
EGFR_1103_1115	P00533	EGFR	GSVQNPVYHNQPL	(1110)	Cluster C	**EGFR**	**EGFR**
VGFR2_944_956	P35968	KDR	RFRQGKDYVGAIP	(951)	Cluster C	**VEGFR2**	**VEGFR2**
MK14_173_185	Q16539	MAPK14	RHTDDEMTGYVAT	(182)	Cluster C	**MAP2K3, MAP2K4**, MAP14, **MAP2K6**	**MAP2K3, MAP2K4, MAP2K6**, MAP3K6
EPHA4_589_601	P54764	EPHA4	LNQGVRTYVDPFT	(596)	Cluster C	**EPHA4**	**EPHA4**
MK01_180_192	P28482	MAPK1	HTGFLTEYVATRW	(187)	Cluster C	**MAP2K1, MAP2K2**	JAK2, EGFR, **MAP2K2**, Ret, **MAP2K1**

Bold characters indicate predicted kinases using two database analysis including UniProt and Phosphosite Plus.

**Table 3 proteomes-11-00006-t003:** Pathway analysis of kinase activity profiles in NSCLC patients with sensitizing EGFR mutations.

#term ID	Term Description	Observed Gene Count	Background Gene Count	Strength	False Discovery Rate
hsa04151	PI3K-Akt signaling pathway	26	350	1.27	4.27 × 10 ^−23^
hsa04014	Ras signaling pathway	21	226	1.37	2.53 × 10 ^−20^
hsa04015	Rap1 signaling pathway	20	202	1.4	6.04 × 10 ^−20^
hsa05200	Pathways in cancer	26	517	1.1	1.29 × 10 ^−19^
hsa04010	MAPK signaling pathway	21	288	1.26	1.12 × 10 ^−18^
hsa05205	Proteoglycans in cancer	17	196	1.34	4.08 × 10 ^−16^
hsa01521	EGFR tyrosine kinase inhibitor resistance	13	78	1.62	1.54 × 10 ^−15^
hsa05235	PD-L1 expression and PD-1 checkpoint pathway in cancer	13	88	1.57	5.55 × 10 ^−15^
hsa04510	Focal adhesion	16	198	1.31	7.92 × 10 ^−15^
hsa04360	Axon guidance	15	177	1.33	3.50 × 10 ^−14^
hsa05215	Prostate cancer	12	96	1.5	3.99 × 10 ^−13^
hsa05230	Central carbon metabolism in cancer	11	69	1.6	4.38 × 10 ^−13^
hsa04660	T cell receptor signaling pathway	12	101	1.47	5.90 × 10 ^−13^
hsa04722	Neurotrophin signaling pathway	12	114	1.42	2.07 × 10 ^−12^
hsa04012	ErbB signaling pathway	11	83	1.52	2.21 × 10 ^−12^
hsa04658	Th1 and Th2 cell differentiation	11	87	1.5	3.32 × 10 ^−12^
hsa05167	Kaposi sarcoma-associated herpesvirus infection	13	187	1.24	2.08 × 10 ^−11^
hsa04670	Leukocyte transendothelial migration	11	109	1.4	2.85 × 10 ^−11^
hsa05161	Hepatitis B	12	159	1.28	5.99 × 10 ^−11^
hsa05135	Yersinia infection	11	125	1.34	1.02 × 10 ^−10^

**Table 4 proteomes-11-00006-t004:** Hub-bottlenecks of the NSCLC network based on comprehensive kinase activity data.

ID	Kinase	Uniprot ID	Betweenness Centrality	Closeness Centrality	Degree	Stress
1	CTNNB1	P35222	0.180455356	0.768421053	51	4522
2	EGFR	P00533	0.100566651	0.760416667	51	3226
3	PIK3R1	P27986	0.061532117	0.688679245	42	1996
4	ERBB2	P04626	0.043504506	0.688679245	41	1742
5	PTPN11	Q06124	0.036477003	0.688679245	41	1594
6	PLCG1	P19174	0.04424585	0.651785714	39	1566
7	CRK	P46108	0.03378544	0.646017699	36	1354

**Table 5 proteomes-11-00006-t005:** A list of peptides and kinases related to prognosis in NSCLC patients with sensitizing EGFR mutations.

Peptide Site	Phosphorylation Site	Signal Intensity	Fold Change	*p* Value	FDR q Value	PhosphoPlus	Uniprot
DYR1A_312_324	(19, 321)	13.52805	3.116532	9.83 × 10 ^−8^	3.34 × 10 ^−6^	EGFR	-
VINC_815_827	(822)	6.042773	3.077846	1.55 × 10 ^−5^	0.000144	KARS	-
ACHD_383_395	(383, 390)	6.138561	2.551585	0.00015	0.001174	-	-
MK01_180_192	(187)	9.516106	2.024014	5.80 × 10 ^−6^	6.58 × 10 ^−5^	EGFR	MAP2K1, MAP2K2
CD3Z_116_128	(123)	42.36624	1.731	0.001188	0.006731	-	-
EFS_246_258	(253)	1559.662	1.719715	0.013831	0.04551	-	Src
ANXA2_17_29	(24)	20.03061	1.659909	0.009685	0.036588	Src, IGF1R	Src
EGFR_1103_1115	(1110)	21.06897	1.621966	0.01073	0.03774	**EGFR**	**EGFR**
EPOR_419_431	(368)	59.07756	1.599058	0.001489	0.007994	**JAK2**	**JAK2**
PRGR_786_798	(795)	14.26506	1.579949	0.002504	0.012161	-	-
SRC8_CHICK_476_488	(477, 483)	1262.13	1.566668	1.80 × 10 ^−6^	3.01 × 10 ^−5^	-	Src
P85A_600_612	(607)	419.0784	1.515741	0.000528	0.00359	EGFR, INSR, CSFR	-

Bold characters indicate predicted kinases using two database analysis including UniProt and Phosphosite Plus.

**Table 6 proteomes-11-00006-t006:** Pathway analysis for NSCLC patients harboring sensitizing EGFR mutations with poor prognosis.

#Term ID	Term Description	Observed Gene Count	Background Gene Count	Strength	False Discovery Rate
hsa04151	PI3K-Akt signaling pathway	26	348	1.27	2.74 × 10 ^−23^
hsa04014	Ras signaling pathway	20	228	1.34	5.64 × 10 ^−19^
hsa04015	Rap1 signaling pathway	19	203	1.37	1.28 × 10 ^−18^
hsa04010	MAPK signaling pathway	21	293	1.25	1.32 × 10 ^−18^
hsa05200	Pathways in cancer	25	515	1.08	1.32 × 10 ^−18^
hsa05205	Proteoglycans in cancer	18	195	1.36	8.99 × 10 ^−18^
hsa04510	Focal adhesion	17	197	1.33	2.40 × 10 ^−16^
hsa01521	EGFR tyrosine kinase inhibitor resistance	13	78	1.62	8.07 × 10 ^−16^
hsa04360	Axon guidance	15	173	1.33	1.74 × 10 ^−14^
hsa05230	Central carbon metabolism in cancer	11	65	1.62	1.69 × 10 ^−13^
hsa05215	Prostate cancer	12	97	1.49	2.64 × 10 ^−13^
hsa04670	Leukocyte transendothelial migration	12	112	1.42	1.17 × 10 ^−12^
hsa04012	ErbB signaling pathway	11	83	1.52	1.48 × 10 ^−12^
hsa04658	Th1 and Th2 cell differentiation	11	88	1.49	2.47 × 10 ^−12^
hsa04660	T cell receptor signaling pathway	11	99	1.44	7.54 × 10 ^−12^
hsa04722	Neurotrophic signaling pathway	11	116	1.37	3.49 × 10 ^−11^
hsa04380	Osteoclast differentiation	11	124	1.34	6.45 × 10 ^−11^
hsa04370	VEGF signaling pathway	9	59	1.58	6.57 × 10 ^−11^
hsa04068	FoxO signaling pathway	11	130	1.32	9.30 × 10 ^−11^
hsa01522	Endocrine resistance	10	95	1.42	1.10 × 10 ^−10^

**Table 7 proteomes-11-00006-t007:** Network analysis of kinases based on prognosis.

Kinase	Uniprot ID	Degree	Betweenness Centrality	Closeness Centrality	Stress
EGFR	P00533	19	0.230237725	0.851851852	306
PIK3R1	P27986	17	0.183294549	0.793103448	248
ERBB2	P04626	17	0.120685682	0.793103448	218

## Data Availability

Not applicable.

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
