# Peer review of "Comprehensive Kinase Activity Profiling Revealed the Kinase Activity Patterns Associated with the Effects of EGFR Tyrosine Kinase Inhibitor Therapy in Advanced Non-Small-Cell Lung Cancer Patients with Sensitizing EGFR Mutations"

_proteomes, 2023, doi:10.3390/proteomes11010006_

Round 1
Reviewer 1 Report
The authors collected 19 stage IV NSCLC patients with sensitizing EGFR mutations, performed comprehensive kinase activity profiling and analyzed in combination with the prognoses after the administration of EGFR-TKIs. They found patients with poor prognosis exhibited high activation of EGFR, PIK3R1, and ERBB2. This research may provide predictive biomarker candidates for screening patients with advanced NSCLC harboring sensitizing EGFR mutations.
However, there is one major concern:
In Figure 2A, the profile of first three samples of cluster 2 is more similar to cluster 1 other than 2, the authors should clarify the influence to their findings.
Author Response
Comment
In Figure 2A, the profile of first three samples of cluster 2 is more similar to cluster 1 other than 2, the authors should clarify the influence to their findings.
Response to the comment
We truly appreciated the comment. We are aware that the three samples, which is now in cluster 2, was more similar to cluster 1 than the cluster 2. We looked into the patient’s characteristics including prognostic events, but we couldn’t find the factor that would make those samples more similar to cluster 1 other than cluster 2. We mentioned about this observation in the discussion of the revised manuscript (p.15 line 410-413).
Reviewer 2 Report
Noguchi and co-workers present a comprehensive kinase activity in advanced NSCLC patients with sensitizing EGFR muattions treated with EGFR TK inhibitors. They can identify two clusters with highly activated kinases and, according to the Authors, one cluster with activated EGFR, HER2 and PIK3R1 is associated to a worse outcome. The data are of interest and hypothesis-generating but certainly not conclusive for many reasons (some already acknowkedged by the authors in the discussion): limited sample size (18 patients only), overall a good outcome, many confounding factors not taken into considerations (i.e: different efficacy of osimertinib vs other EGFR TKIs; site of metastases and tumor burden, PS etc). Also the title may be misleading ( "....predicts effects of EGFR tyrosine kinease inhibitors...) and should be mitigated.
Author Response
Comment 1
The data are of interest and hypothesis-generating but certainly not conclusive for many reasons (some already acknowledged by the authors in the discussion): limited sample size (18 patients only), overall a good outcome, many confounding factors not taken into considerations (i.e: different efficacy of osimertinib vs other EGFR TKIs; site of metastases and tumor burden, PS etc).
Response to the comment 1
We thank the reviewer for the constrictive comment to the limitation of our study. We agreed with the pointed out limitations, and we discussed about them in the revised manuscript (p.15 line 405-420).
Comment 2
The title may be misleading ( "....predicts effects of EGFR tyrosine kinease inhibitors...) and should be mitigated.
Response to the comment 2
We found that the title may be misleading and should be mitigated. According to the reviewer’s suggestion, we have changed the title as “Comprehensive kinase activity profiling revealed the kinase activity patterns associated with effects of EGFR tyrosine kinase inhibitors therapy in advanced non-small cell lung cancer patients with sensitizing EGFR mutations”.